# Antibiogram of Urinary Tract Infections and Sepsis among Infants in Neonatal Intensive Care Unit

**DOI:** 10.3390/children9050629

**Published:** 2022-04-28

**Authors:** Abdulrahman S. Bazaid, Abdu Aldarhami, Hattan Gattan, Heba Barnawi, Husam Qanash, Ghaida Alsaif, Bandar Alharbi, Abdulaziz Alrashidi, Essam Hassan Eldrehmy

**Affiliations:** 1Department of Medical Laboratory Science, College of Applied Medical Sciences, University of Ha’il, Hail 55476, Saudi Arabia; h.barnawi@uoh.edu.sa (H.B.); h.qanash@uoh.edu.sa (H.Q.); g.alsaif@uoh.edu.sa (G.A.); b.alharbi@uoh.edu.sa (B.A.); am.alrashidi@uoh.edu.sa (A.A.); 2Department of Medical Microbiology, Qunfudah Faculty of Medicine, Umm Al-Qura University, Al-Qunfudah 21961, Saudi Arabia; ahdarhami@uqu.edu.sa; 3Department of Medical Laboratory Technology, Faculty of Applied Medical Sciences, King Abdulaziz University, Jeddah 21589, Saudi Arabia; hsqattan@kau.edu.sa; 4Special Infectious Agents Unit, King Fahad Medical Research Center, Jeddah 22252, Saudi Arabia; 5Molecular Diagnostics and Personalized Therapeutics Unit, University of Ha’il, Hail 55476, Saudi Arabia; 6Department of Biology, Turabah University College, Taif University, Turabah 21995, Saudi Arabia; esam2005micro@gmail.com; 7Department of Microbiology, Faculty of Veterinary Medicine, Zagazig University, Zagazig 44519, Egypt

**Keywords:** sepsis, antimicrobial resistance, neonatal intensive care unit, urinary tract infections

## Abstract

Neonatal infections including sepsis and urinary tract infections are considered among the leading causes of mortality in neonatal intensive care units (NICU). Thus, use of empiric antibiotics is very important in infected neonates and the success of this practice is mainly reliant on the availability of an up-to-date antibiogram for currently used antibiotic drugs. In this study, we aim to determine the bacteriological profile and antibiotic susceptibility pattern of bacteria isolated from blood or/and urine cultures belonging to patients at the NICU. A total of 54 urine samples were collected in the period between January 2015 and December 2019. Data of infants with positive urine and blood bacterial isolates were gathered retrospectively. The most commonly isolated bacteria from urine observed were *K. pneumoniae* (44%) and *E. coli* (39%), while *Acinetobacter baumannii* (33%) and *K. pneumoniae* (22%) predominated in neonatal blood samples. The majority of uropathogens and blood isolates exhibited low resistance to imipenem and tigecycline, respectively. These antibiotics would be recommended for future use as empirical treatment in neonates with urinary tract infections and/or sepsis. This investigation highlights the importance of surveillance studies to manage and ensure the effectiveness of treatment plan for critically ill infants.

## 1. Introduction

Neonates or newborns can be defined as babies from the time of delivery up to four weeks old [1]. Neonates possess an incompetent innate or/and adaptive immunity, which make them more susceptible and less able to compete with infections caused by numerous pathogenic microorganisms [2]. Bacterial infections in newborns can range from mild to severe and life-threating infections, including urinary tract (UTIs) and blood stream infections (BSIs) [3]. Based on the age of neonates when contracting the infection, both BSIs and UTIs can be categorised into two groups: early (during the first 7 days of life) and late (after 7 days from delivery) onset [4]. This is because the immune system of neonates is rapidly developing as they are growing, thus each stage of age may possess different level of potency to fight infections [4]. This fact might explain the reasons behind observed variation of causative bacteria according to baby’s age group [2].

UTIs are very common among neonates, which can be defined as pyuria in urinalysis and a urine culture with 10,000–50,000 colony forming unit (CFU)/mL [5]. Although multiple previous studies indicated the difficulty of conducting urine test for newborns, such a test can be very informative and an alarming sign for serious bacterial infection, especially sepsis [5,6]. *Escherichia coli*, group B *Streptococcus* (GBS) and *Staphylococcus aureus* are well-associated with early and late onset bacterial infections among neonates [7]. Nevertheless, bacterial infections in neonates with vesicoureteral reflux (VUR) are found to be predominately caused by *Enterobacter aerogenes*, *Klebsiella pneumoniae*, *Proteus mirabilis*, *Proteus vulgaris* and *Pseudomonas aeruginosa* [5].

The presence of bacteria in the blood without multiplication and production of its toxins is called bacteraemia [8]. However, sepsis can be generally identified by the presence of a proliferating microorganism (e.g., bacteria, virus, parasite, or fungi) or/and their produced toxin/s in the patients’ blood stream leading to haemodynamic changes and clinical manifestations of sepsis [9], which will be the focus of this paper. Sepsis is a worldwide cause of morbidity and mortality in the neonatal intensive care units (NICU) [10]. Moreover, the most common cause of the early onset sepsis in neonates are GBS, *Streptococcus agalactiae* and *E. coli*, while the late onset sepsis is caused by GBS, *E. coli*, *Klebsiella* species *Pseudomonas* species, *Acinetobacter* species and *S. aureus* [11]. 

Absence of effective drugs because antimicrobial resistance (AMR) is an unfortunate fact behind the high level of severity of bacterial infections among immunosuppressed individuals, including neonates [12]. It was reported that Gram-negative bacteria (e.g., *E. coli*, *Enterobacter* species, *Klebsiella* species and *Pseudomonas* species) are well linked with neonates’ infections presented with high level of resistance towards ampicillin, amikacin and gentamycin [13]. Another study revealed a high level of resistance of Gram-negative bacteria against cefotaxime, ceftazidime, amikacin and ciprofloxacin leading to almost four times increasing rate of mortality among infected neonates [14]. 

In the early onset sepsis, babies could contract the infection before or during the delivery through ascending infection caused by microbial colonization of the maternal perineum or by the direct contact with the causative microorganism during the delivery [15]. Mothers’ vaginal bacterial microbiota were found to be the most common pathogens associated with the early onset bacterial infections among newborns, including sepsis [15]. However, in the late onset sepsis, the source of the infection could be nosocomial [16]. Early and late sepsis in neonates could be caused by multiple Gram-positive and negative bacteria, including coagulase-negative staphylococci (CoNS), *E. coli*, *Enterococci*, *Haemophilus influenza*, *Listeria monocytogenes*, *Pseudomonas aeruginosa*, *S. aureus* and *Streptococcus pyogenes* [16,17]. Generally, septic pathogens or bacteria is commonly associated with certain infections, including meningitis, pneumonia, arthritis, osteomyelitis, phlebitis, gastroenteritis and UTIs [18]. Thus, it is a very rational practice that clinicians screen for sepsis in patients with these illnesses and vice versa, including UTIs and BSIs.

As the immune system of neonates is not very competent, bacterial infections in this group of patients would require very special care and a treatment plan to avoid any serious complications [2]. Additionally, the increase level of resistance against antibiotic drugs used to treat bacterial infections associated with sepsis and UTI in neonates is very alarming worldwide [19]. Furthermore, use of empirical antibiotics is very important in infected neonates, especially in septic patients, and the success of this kind of practice would mainly relay on the availability of a recent surveillance (antibiogram) for antibiotic resistance nationally and internationally.

The primary goal of antimicrobial stewardship (AMS) is to improve the effective use of antibiotics in different sectors (human, animals, agriculture and aquaculture) worldwide [20]. AMS in human health, especially in hospitals, can be applied by different means, including the availability of antibiogram that is easily accessible by all local physicians [20]. This is because antibiogram would provide doctors with a rational guideline when prescribing empirical antibiotics to treat bacterial infections, particularly life-threating infections such as sepsis [21]. Previous local antibiogram studies have contributed to the development or alteration of first line treatments in numerous scenarios [21]. For instance, the serious challenge of emerging carbapenem-resistant *P. aeruginosa* in a USA hospital was found to be associated with co-resistance to first-line treatments (beta-lactams) and delayed application of effective empiric antibiotics, thus urgent modification of used antibiotics was carried out [22]. Nevertheless, this observation was only obtained after conducting a local antibiogram study for this bacterium, and this should highlight the importance of antibiogram investigation. Consequently, it is crucial to incorporate an antibiogram into clinical decision to achieve the main goal of providing effective empiric antibiotic therapy. The aim of this project, therefore, is to assess the prevalence of UTIs and its correlations with BSIs among neonates of Hail region in the Kingdom of Saudi Arabia as well as providing recent surveillance data of antibiotics resistance in causative bacteria of these infections. 

## 2. Materials and Methods

### 2.1. Study Design

A retrospective study was conducted in neonatal intensive care unit (NICU) at Hail Maternity Hospital over a period of five years (January 2015 to December 2019). Medical data of neonates admitted to the NICU (up to 30 days old) with a positive urine culture were obtained from the clinical microbiology laboratory database. Data of neonates with positive blood culture were collected only for those with UTIs to study any potential correlation between UTIs and BSI.

### 2.2. Study Population

Urine and blood cultures are requested simultaneously by the local clinical doctors for neonates presented with signs and symptoms of septicaemia, including fever (>38 °C), hypothermia (<36 °C), metabolic acidosis, or tachypnoea [23] in the absence of any antibiotic therapy. Urine was collected using short-term urinary catheters (in-and-out catheterisation). Positive urine cultures were identified when the colony forming units (CFU) of a single detected bacterial species were ≥10^5^ CFU/mL [24]. Cultures with polymicrobial growth (>one species) or low number of colonies (<10^5^ CFU/mL) were excluded from this investigation. Blood samples were collected aseptically into an appropriate BACTEC bottle (BD BACTEC™) and incubated at 37 °C using the incubator of the blood culture system (BACTEC-9050, Beckton-Dickenson, Franklin Lakes, NJ, USA) [25].

### 2.3. Bacterial Identification and Antibiotic Sensitivity Testing

Bacterial identification and antibiotic sensitivity testing were conducted using the VITEK system (bioMerieux, Hampshire, UK) [26]. Obtained data were analysed and interpreted according to the guidelines of the Clinical Laboratory Standards Institute (CLSI) as explained previously [24]. Susceptibility of recovered Gram-negative UTI-related bacteria was evaluated against a panel of antibiotics, including amikacin, ampicillin, cefepime, ceftazidime, cefoxitin, ciprofloxacin, gentamicin, imipenem, meropenem, nitrofurantoin, piperacillin, tigecycline and trimethoprim/sulfamethoxazole. For Gram-positive UTI-associated bacteria, assessment was conducted against ampicillin, cephalothin, clindamycin, levofloxacin, linezolid, moxifloxacin, nitrofurantoin, penicillin, tigecycline, trimethoprim/sulfamethoxazole and vancomycin. The sensitivity testing of ceftazidime against Gram-negative rods was especially conducted to determine extended-spectrum β-lactamase (ESBL) producing strains [27].

### 2.4. Statistical Analysis

Patients ‘gender, urine and blood culture reports and antibiotic resistance profiles were electronically retrieved from the hospitals’ medical records, and the data were presented as the numbers of cases for each isolated bacterium. Statistical analysis was performed using SPSS software version 16.0 (IBM Corporation, Armonk, NY, USA). 

## 3. Results

### 3.1. Prevalence of Uropathogens and Septic Bacteria in NICU

A total of 54 (12.6%) of all positive urine cultures during the study period (January 2015 to December 2019) belong to NICU. Males’ neonates account for 66.7% of their urine culture, while the remained portion (33.3%) belong to female neonates. Males were predominant in all five years except in 2015, females were higher than males (9 and 7, respectively). About 9 out of 54 (16.6%) neonates with positive urine culture were identified with bacteraemia. Prevalence of urine associated bacteria among NICU was determined as follows: *K. pneumoniae* (44%), *E. coli* (39%), *Enterococcus faecalis* (7.5%), *Pseudomonas aeruginosa* (5.6%) and *Streptococcus agalactiae* (3.8%). In addition, the dissemination of bacteria among septic NICU was calculated as follows: *Acinetobacter baumannii* (33.3%), *K. pneumoniae* (22.2%), *E. coli* (11.1%), *P. aeruginosa* (11.1%), *Staphylococcus aureus* (11.1%) and *Streptococcus pneumoniae* (11.1%).

### 3.2. Simultaneous Infection/Colonisation of Identified Bacterial Isolates in Blood and Urine of NICU Patients with Positive Blood Cultures

To determine any potential correlation between bacteria identified simultaneously from blood and urine of NICU, data of all neonates with sepsis and positive urine cultures were determined (Table 1). The majority (77%) of identified bacteria in the blood of NICU patients were Gram-negative bacteria (Table 1). Likewise, in urine specimens, about 88% of identified uropathogens in patients with sepsis are Gram-negative bacteria. Nevertheless, about 66% of all Gram-negative uropathogens in neonates with bacteraemia belong to *K. pneumoniae* (Table 1). Two cases out of 9 septic neonates were identified with identical bacterial species in both specimens (blood and urine) (Table 1).

### 3.3. Sensitivity Profiles of Identified Gram-Negative Bacteria from Blood and Urine of NICU Patients

Susceptibility profiles were determined for isolates recovered from urine and/or blood specimens using 12 commonly used antibiotics in the local hospital. Urine isolates of non-ESBL producing strains of *E. coli* and *K. pneumoniae* showed high percentage of resistance to piperacillin and trimethoprim/sulfamethoxazole, whereas wide sensitivity (zero resistance) of these isolates towards cefoxitin was also monitored (Table 2). In addition, a high portion of *K. pneumoniae* ESBL-producing strains isolated from urine and/or blood cultures of neonates exhibited resistance to ampicillin, cefepime and gentamicin, while absent resistance by the same strains was detected when using imipenem, meropenem and tigecycline (Table 2). About 85% of *K. pneumoniae* ESBL-producing showed resistance to cefepime, while the same drug was completely active with zero resistance towards non-ESBL producing strains of this bacterium (Table 2). Furthermore, the majority of tested antibiotics showed limited activity against ESBL and non-ESBL *Pseudomonas aeruginosa* recovered from urine and/or blood cultures of neonates except piperacillin as its broad activity towards all strains of *P. aeruginosa* was observed (Table 2). *Acinetobacter baumannii* that was isolated from blood of NICU were totally resistant to all tested antibiotics, except tigecycline only (Table 2). 

### 3.4. Sensitivity Profiles of Identified Gram-Positive Bacteria from Blood and Urine of NICU Patients

A quarter and half of *Enterococcus faecalis* and *Streptococcus agalactiae* isolated from urine were resistant to clindamycin, nitrofurantoin and trimethoprim/sulfamethoxazole, respectively (Table 3). Blood isolates of methicillin-sensitive *Staphylococcus aureus* (MSSSA) were susceptible to almost all tested antibiotics, excluding trimethoprim/sulfamethoxazole. On the other hand, blood associated *Streptococcus pneumoniae* exhibited a high level of resistance to all used antibiotics apart from vancomycin and tigecycline (Table 3).

## 4. Discussion

There are multiple factors that have been linked with the emergence and spread of multidrug resistant bacteria (MDR) worldwide, including unguided use/prescribing of antibiotic drugs [28]. Infection with drug resistant bacteria carries a high rate of mortality, especially among immune compromised individuals, such as newborns [28]. Prevalence of UTI amongst neonates varies between studies which ranged from 7% to 15% based on the sample size and location of the study [29]. Thus, this study was designed to identify the uropathogens and potential association with septic bacteria among NICU in Hail region, Saudi Arabia, between 2015 and 2019. 

The current investigation includes patients who were admitted to the NICU that made about 12.6% (54) of all positive urine cultures during the study period. The majority of neonatal positive urine cultures belong to male patients in all five years, except 2015. Among these neonates with UTIs, sepsis was identified in 9 of 54 (16.6%) newborns. This finding would lead to the suggestion that male newborns are more prone to contract UTIs than females. This observation was backed up by a prior similar study reporting that the prevalence of UTIs among boys was higher than girls with the following calculated ratio: (4.5:1) [30]. Generally, UTIs tend to be more frequent in females than in male individuals across all age groups based on the actual differences in the length of their urethra and anatomical differences in males [31], and this would be the possible reason behind the higher rate of UTIs in females compared to males. However, this was not the case when referring to uncircumcised males as multiple studies have revealed their high vulnerability to contracting UTIs compared with their circumcised counterparts [32]. Male are predominant (90%) in neonates with UTI aged less than 30 days in a ten years study [26]. Uncircumcised infant boys aged less than 60 days have had higher rate of UTIs (21%), compared to their circumcised counterparts (2%) and girls (5%) [5]. This fact might explain the high percentage (66%) of UTIs in male neonates as they may not undergo any circumcision at this stage.

It was claimed that neonates with positive urine cultures would possess a higher probability to develop sepsis [33]. About third of neonates who were diagnosed with UTI presented with positive blood cultures and sepsis [33]. Currently, about 16% of neonates have had urosepsis, and this would agree with a report from China where 11% of neonates with UTI have had septicaemia [34]. Neonates admitted to the NICU are usually considered as critically ill patients, which means the use of central lines could be a necessity, although use of central lines is strongly linked with an increased risk of BSIs [35]. Therefore, high prevalence of septicaemia are reported among NICU where active antibiotic should be implemented immediately [36].

Generally, Gram-negative bacteria are more prevalent in urine and blood specimens of patients in the NICU compared to Gram-positive bacteria. *Klebsiella pneumoniae* and *E. coli* were the top identified bacteria from urine cultures of neonates, while *Acinetobacter baumannii* was observed more often in septic neonates. In line with previous reports, *Klebsiella* species and *E. coli* were found to be the bacteria most commonly associated with infections of newborns [37,38,39]. In addition, a big portion of the UTI infections in neonates are due to *K. pneumoniae* and *E. coli*, which were more linked with males than females [40]. Moreover, the majority (77%) of detected bacteria from blood samples of patients at the NICU were Gram-negative bacteria and roughly 88 percent of identified uropathogens in septic patients. Nevertheless, *K. pneumoniae* is responsible for around two thirds of Gram-negative uropathogens in neonates with sepsis, and only 22% of BSIs were caused by Gram-positive bacteria, including *Staphylococcus aureus* and *Streptococcus pneumoniae*.

In paediatric UTI, young age is a well-known risk factor for sepsis [41]. Previous study demonstrated that *E. coli* and *K. pneumoniae* were the two most common pathogens (81.2%) in UTIs [42]. According to other studies, *E. coli* is the most common cause of UTIs in neonates [43]. However, certain studies have shown that the total burden of disease caused by *E. coli* was lower in young age group (approximately 50% of all positive cultures) than in older age groups, in which *E. coli* was responsible for up to 80% of UTIs. Male newborns with VUR were more likely than female infants to have UTIs caused by *K. pneumoniae*, *K. oxytoca*, *Proteus mirabilis*, *Proteus vulgaris*, *Enterobacter aerogenes* and *Pseudomonas aeruginosa* [26,44]. Although occurrences of *Enterococcus faecalis*, *Staphylococcus aureus*, *Group B Streptococcus* and *Streptococcus pneumonia* infections in newborns are uncommon, cases of *Enterococcus faecalis*, *Staphylococcus aureus*, *Group B Streptococcus* and *Streptococcus pneumonia* have been reported [45].

Sensitivity profiles of non-extended spectrum β-lactamase (non-ESBL) *K. pneumoniae* strains that were detected from urine samples of neonates showed high percentages of resistance to piperacillin and trimethoprim/sulfamethoxazole, whereas the high sensitivity of these isolates to cefoxitin was monitored. Meanwhile, *K. pneumoniae* ESBL strains were resistant to ampicillin, cefepime and gentamicin, except imipenem, meropenem and/or tigecycline. In line with Pokhrel et al. [46], the majority of ESBL producing *K. pneumoniae* (85%) were resistant to cefepime. Additionally, a high resistance rate was reported for *K. pneumoniae* against trimethoprim/sulfamethoxazole followed by piperacillin, cefotaxime, cefepime and aztreonam, while colistin drug showed a high level of activity towards the same isolates [47].

Culturing of blood samples revealed that the most common bacterial isolate was *K. pneumoniae* and that the majority of isolates were highly resistant to gentamicin and ampicillin (Table 1). *Escherichia coli* isolates have been reported with high sensitivity to ampicillin and cefepime in the Hail region [24]. A combination therapy made of gentamicin and ampicillin would be recommended as empirical treatment for neonates with UTI or/and sepsis in the local hospital. Thus, this study highlights the importance of antibiogram to ensure the effectiveness of therapeutic drugs.

Nonetheless, previous neonatal research had identified coagulase negative staphylococcus aureus (CoNs) as predominant bacterial species associated with blood specimens [48]. Variation among location of studies, size of recruited patients and level of adherence to hand hygiene techniques would explain the difference in the prevalence of bacteria associated with neonatal infections.

Concerning the sensitivity profile, *A. baumannii* was resistant to all tested antibiotics except for tigecycline. The resistance profile of *A. baumannii* was claimed by Shaw et al. [49] although another study indicated the high sensitivity of tested *A. baumannii* strains to ciprofloxacin and colistin as well as tigecycline [46]. Tigecycline was successfully used previously to treat extensively drug-resistant *Klebsiella pneumoniae* in septic neonates [50]. Additionally, although tigecycline was effectively applied to eradicate extensively drug-resistant *Acinetobacter baumannii* in infected neonates, usage of tigecycline in this group of patients in relation to its safety might be questionable [51]. However, using tigecycline in neonates with life-threating infections such as sepsis, especially where an alternative, safer and active therapeutic agent is missing, was justified [51]. Thus, the current antibiogram would propose the local potential use of tigecycline as an empirical drug in neonates with sepsis caused by *Acinetobacter baumannii*. Nevertheless, this data may not be representative of another region within the kingdom or worldwide. This is due to the observed differences among the sensitivity profiles of *A. baumannii* strains from one location to another, which may attributed to the variation of therapeutic drugs used locally for treating this bacterium. Multiple strains of *Pseudomonas aeruginosa* were tested against 12 antibiotics, all of which were sensitive to imipenem and meropenem. A previous study presented similar data to the current findings in which eight isolates were tested against the antibiotics used in this study, showing resistance to ciprofloxacin and tigecycline [52]. Multiple former studies have reported the high sensitivity of *Enterococcus faecalis* towards vancomycin [53]. *Enterococcus faecalis* and *Streptococcus agalactiae* were also assessed in the present study where they were resistant to clindamycin, nitrofurantoin and trimethoprim/sulfamethoxazole. This observation is in agreement with Karthikeyan et al., who reported that all tested isolates of *Enterococcus faecalis* and *Streptococcus agalactiae* were sensitive to vancomycin (100%) [54].

The results of the current antibiogram are an essential tool for local hospitals and policy makers in Saudi Arabia to track bacterial resistance and change their choice of empirical drugs accordingly to ensure their activity and thus control the spread of AMR. In addition, sensitivity profiles among bacterial isolates nationally and worldwide over a short period of time were varied [20], and this would indicate the importance of conducting frequently a local antimicrobial surveillance study. Having local and up-to-date antibiogram data does not necessarily means better prescription of antibiotics and limited spread of AMR, unless antibiogram data are well distributed and utilised by the local health care workers (HCWs). This can be achieved by different means, including a series of targeted meetings for local clinical microbiologists and physicians to raise awareness about the importance of establishing and using local antibiogram data. This is because using local antibiogram effectively could lead to a significant reduction in inappropriate prescription of antibiotics, level of local AMR, rate of local hospitalisation and rate of probability for local patients to develop serious complications from infections and death.

Although this study has assessed the prevalence of UTIs and their correlation with BSIs among infants in NICU, it has several limitations. The number of recruited patients may not be representative of all neonates with different types of infections, since larger populations with wider bacterial infections could lead to better outcomes. In addition, the limited number of Gram-positive bacteria and the absence of *Listeria monocytogenes* could also be considered as potential limitations.

## 5. Conclusions

Gram-negative bacteria were the major cause of UTIs and sepsis among ICU babies. The most common causes of UTIs were *K. pneumoniae* and *E. coli* while sepsis was caused by *Acinetobacter baumannii* and *K. pneumoniae.* Urine and blood isolates showed *K. pneumoniae* and *E. coli* with ESBL and non-ESBL activity. The non-ESBLs were highly resistant to piperacillin and trimethoprim/sulfamethoxazole; on the other hand, the ESBLs were resistant to ampicillin, cefepime and gentamicin. Piperacillin and tigecycline were the only active drug against *P. aeruginosa* and *A. baumannii,* respectively. The majority of uropathogens and blood isolates exhibited low resistance to imipenem and tigecycline, respectively, which can be proposed for potential usage as empirical treatment in neonates with sepsis and UTIs. These data highlight the importance of surveillance studies in managing critically ill infants.

## Figures and Tables

**Table 1 children-09-00629-t001:** Simultaneous infection/colonisation of identified bacterial isolates in blood and urine of neonatal intensive care unit (NICU) patients with positive blood cultures.

Neonates with Sepsis	Identified Bacteria in the Blood	Identified Bacteria in the Urine
Case 1	*Acinetobacter baumannii*	*Klebsiella pneumoniae*
Case 2	*Acinetobacter baumannii*	*Klebsiella pneumoniae*
Case 3	*Acinetobacter baumannii*	*Streptococcus agalactiae*
Case 4	*Escherichia coli **	*Escherichia coli* *
Case 5	*Klebsiella pneumoniae*	*Escherichia coli*
Case 6	*Klebsiella pneumonia **	*Klebsiella pneumoniae* *
Case 7	*Pseudomonas aeruginosa*	*Klebsiella pneumoniae*
Case 8	*Staphylococcus aureus*	*Klebsiella pneumoniae*
Case 9	*Streptococcus pneumoniae*	*Klebsiella pneumoniae*

* Indicates patients identified with identical bacterial species from urine and blood specimens.

**Table 2 children-09-00629-t002:** Resistance profiles of Gram-negative bacteria isolated from positive urine and blood cultures towards a set of commonly used antibiotics in neonatal intensive care unit (NICU).

-	Number of Isolates	Amikacin	Ampicillin	Cefepime	Cefoxitin	Ciprofloxacin	Gentamicin	Imipenem	Nitrofurantoin	Meropenem	Piperacillin	Tigecycline	Trimethoprim/Sulfamethoxazole
Urine isolates													
*Escherichia coli*	21	4 (19)	0	3 (14)	0	7 (33)	7 (33)	4 (19)	5 (24)	3 (14)	9 (43)	6 (29)	13 (62)
*Klebsiella pneumoniae*	17	7 (41)	1 (6)	0	0	2 (12)	7 (41)	0	2 (12)	0	3 (18)	2 (12)	3 (18)
*Klebsiella pneumoniae* ESBL *	7	4 (57)	5 (71)	6 (86)	1 (14)	3 (43)	5 (71)	0	1 (14)	0	1 (14)	0	1 (14)
*Pseudomonas aeruginosa*	3	2 (67)	2 (67)	0	1 (33)	1 (33)	2 (67)	0	2 (67)	0	0	2 (67)	3 (100)
Blood isolates													
*Klebsiella pneumoniae* ESBL *	2	1 (50)	1 (50)	2 (100)	1 (50)	0	1 (50)	0	0	0	0	0	0
*Pseudomonas aeruginosa* ESBL *	1	0	1 (100)	1 (100)	1 (100)	1 (100)	1 (100)	1 (100)	0	1 (100)	0	0	1 (100)
*Acinetobacter baumannii*	3	3 (100)	3 (100)	3 (100)	3 (100)	3 (100)	3 (100)	3 (100)	3 (100)	3 (100)	3 (100)	0	3 (100)

Data are presented as the total number of resistant isolates during the study period. Numbers in brackets are percentages. * Isolates resistant to ceftazidime (third generation cephalosporin) are considered as extended-spectrum β-lactamase producing. Resistance profile of bacterial isolate present in both urine and blood was included among urine isolates only.

**Table 3 children-09-00629-t003:** Resistance profiles of Gram-positive bacteria isolated from neonates with urinary tract infections (UTIs) during a 5-year period (2015–2019) towards a set of commonly used antibiotics.

-	Number of Isolates	Cephalothin	Clindamycin	Levofloxacin	Linezolid	Moxifloxacin	Nitrofurantoin	Penicillin	Tigecycline	Trimethoprim/Sulfamethoxazole	Vancomycin
Urine isolates											
*Enterococcus faecalis*	4	0	1 (25)	0	0	0	1 (25)	0	0	1 (25)	1 (25)
*Streptococcus agalactiae*	2	0	1 (50)	0	0	0	1 (50)	1 (50)	0	1 (50)	0
Blood isolates											
*Staphylococcus aureus*	1	1 (100)	0	0	0	0	0	1 (100)	0	1 (100)	0
*Streptococcus pneumoniae*	1	1 (100)	0	1 (100)	1 (100)	1 (100)	1 (100)	1 (100)	0	1 (100)	0

Data presented as the total number of resistant strains isolated during the study period. Numbers in brackets are percentages.

## Data Availability

All data are available within the manuscript.

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
