# Peer review of "Antibiogram of Urinary Tract Infections and Sepsis among Infants in Neonatal Intensive Care Unit"

_children, 2022, doi:10.3390/children9050629_

Round 1
Reviewer 1 Report
Thank you for your work and your efforts to create and antibiogram for your local NICU, this certainly has local clinical utility. I have a few comments for you to consider if revising this paper:
1) As a physician not practicing at your hospital, I am not sure what you want me to learn or to change about my practice from your article. If the question is just about local choice of antibiotics, that is mostly a local concern. Larger regional data may be important for epidemiology reporting. However, if it about how to use an antibiogram effectively to change the clinical outcome of your patients, that would be more significant and I could apply it to my hospital as well.
2) Revisit you use of the term prophylactic antibiotics- this implies the patient has a risk factor for infection but is not currently sick. We do this for a baby born to a mom with chorioamnionitis for example, or treat mom's with ampicillin prophylactically if they have GBS. If the child has bacteremia or a UTI the antibiotics are therapeutic no prophylactic.
3) Regarding study design you may want to me more clear about current practice- what antibiotics are used for prophylaxis in your NICU, what are used for sepsis, how maybe babies received abx during that study period etc. I was also very confused looking a population of babies with confirmed UTI but then studying blood cultures as well- but not looking at blood culture specimens from other children who did not have a UTI, I was not sure what you were trying to show.
4) regarding conclusions- using your most broad spectrum antibiotics for all patients (prophylactically) is probably not what an antibiotics stewardship program would recommend as you need to reserve those antibiotics for the sickest patients who need them or more resistance will develop.
5)I would recommend your background references focus more on the utility and application of antibiograms and antimicrobial stewardship teams than on the clinical course of these infections. If you describe infections please distinguish between the terms sepsis and bacteremia.
Author Response
Dear Reviewer,
I would first like to thank you very much for the review of our manuscript. The comments were helpful and valuable; the efforts taken to improve our manuscript are really appreciated. A copy of the revised manuscript has been uploaded; also, the response to each point raised has been detailed below.
(1) As a physician not practicing at your hospital, I am not sure what you want me to learn or to change about my practice from your article. If the question is just about local choice of antibiotics, that is mostly a local concern. Larger regional data may be important for epidemiology reporting. However, if it about how to use an antibiogram effectively to change the clinical outcome of your patients, that would be more significant and I could apply it to my hospital as well.
RESPONSE (1): This was addressed in multiple sections of the manuscript and the importance of antibiogram investigation was also added to the discussion to encourage other national and worldwide hospitals to establish and utilize antibiogram.
(2) Revisit you use of the term prophylactic antibiotics- this implies the patient has a risk factor for infection but is not currently sick. We do this for a baby born to a mom with chorioamnionitis for example, or treat mom's with ampicillin prophylactically if they have GBS. If the child has bacteremia or a UTI the antibiotics are therapeutic no prophylactic.
RESPONSE (2): We totally agree with this comment. Thus, the term ‘’prophylactic’’ was replaced throughout the manuscript with the term ‘’empirical’’ as all neonates are already infected, and drugs are therapeutic not preventive.
(3) Regarding study design you may want to me more clear about current practice- what antibiotics are used for prophylaxis in your NICU, what are used for sepsis, how maybe babies received abx during that study period etc. I was also very confused looking a population of babies with confirmed UTI but then studying blood cultures as well- but not looking at blood culture specimens from other children who did not have a UTI, I was not sure what you were trying to show.
RESPONSE (3): This was addressed, and the aim of study was amended to the ‘’prevalence of UTIs and its correlations with BSIs in neonates of Hail region in the Kindem of Saudi Arabia’’ as well as providing recent antibiogram data for urine and/or blood associated bacteria.
(4) regarding conclusions- using your most broad spectrum antibiotics for all patients (prophylactically) is probably not what an antibiotics stewardship program would recommend as you need to reserve those antibiotics for the sickest patients who need them or more resistance will develop.
RESPONSE (4): We completely agreed with your comment, so few lines about tigecycline usage in neonates were added in the discussion section, also conclusion was amended. We proposed the potential use of tigecycline empirically not prophylactically (we changed this now) in neonates with sepsis as his is life-threating infection and should be used when not alterative is available to save patients’ life.
(5) would recommend your background references focus more on the utility and application of antibiograms and antimicrobial stewardship teams than on the clinical course of these infections. If you describe infections please distinguish between the terms sepsis and bacteremia.
RESPONSE (5): Although the focus of our manuscript on septic patients who have diagnosed by clinicians with sepsis and this was mentioned in the method section, we have added few lines about infection and the main difference between bacteremia and sepsis in the introduction.
Reviewer 2 Report
The article focuses on the very serious problem of multi-drug resistant bacteria and the treatment of infectious diseases caused by them. This is an up-to-date topic, especially in patients hospitalized in Intensive Care Units. The aim of this paper is to determine the bacteriological profile and antibiotic susceptibility pattern of blood and urine cultures isolated from neonates hospitalized in NICU. The findings might be helpful in clinical settings.
There are some parts that need rephrasing and correction, and also some limitations that should be underlined.
Authors use the term "prophylaxis". Prophylaxis refers to a state when we do not have a disease, and actions are taken to prevent its onset. Antibiotic use in infected patients is a treatment, not prophylaxis. The authors should rephrase:
- line 20 "use of prophylactic antibiotics in infected neonates"
- line 30 "prophylaxis in neonates with sepsis and urinary tract infection"
- lines 89-90 and 300-301
The materials and method section should be improved:
- the study is rather a retrospective analysis of clinical data than a retrospective cohort study,
- study cohort - is not an adequate term,
- urine cultures obtained with the use of sterile urine bags - this method has a high risk of contamination and false-positive results. The use of an adhesive bag is not a valid enough method for urine culture collection. The reference method for urine collection in infants who are incontinent is urethral catheterization, and this limitation of the study should be underlined.
In the discussion:
While referring to the observation, that males neonates are more often diagnosed with UTIs, the authors should also add, that this preponderance reflects the higher incidence of urinary tract anomalies in males.
Line 216: "neonates who aged less than 60 days" - these are infants, not neonates.
I would suggest adding a paragraph about the limitations of the study, e.c. urine collection method, a small number of identified Gram-positive bacteria, etc.
English editing is required. For example:
- line 44 - "as they growing"
- lines 89-80 - "important is infected neonates"
- lines 100-102- should be rephrased
References:
most of the cited articles were published within the last 20 years, however, the authors cite articles referring to urinary tract infections in infants published in 1993, 1993, and 1982. I would suggest looking for more recently published articles.
Author Response
Dear Reviewer,
I would first like to thank you very much for the review of our manuscript. The comments were helpful and valuable; the efforts taken to improve our manuscript are really appreciated. A copy of the revised manuscript has been uploaded; also, the response to each point raised has been detailed below.
Authors use the term "prophylaxis". Prophylaxis refers to a state when we do not have a disease, and actions are taken to prevent its onset. Antibiotic use in infected patients is a treatment, not prophylaxis. The authors should rephrase:
- line 20 "use of prophylactic antibiotics in infected neonates"
- line 30 "prophylaxis in neonates with sepsis and urinary tract infection"
- lines 89-90 and 300-301
RESPONSE: We totally agree with this comment. Thus, the term ‘’prophylactic’’ was replaced throughout the manuscript with the term ‘’empirical’’ as all neonates are already infected, and drugs are therapeutic not preventive.
The materials and method section should be improved:
- the study is rather a retrospective analysis of clinical data than a retrospective cohort study,
- study cohort - is not an adequate term,
RESPONSE: The reference method for urine collection in infants who are incontinent is urethral catheterization. Urine specimens were collected from neonates using short-term urinary catheters (in-and-out catheterisation). This information was amended in the method section now after re-check with the NICU ward to short-term urinary catheters.
In the discussion:
While referring to the observation, that males neonates are more often diagnosed with UTIs, the authors should also add, that this preponderance reflects the higher incidence of urinary tract anomalies in males.
RESPONSE: This was added to the discussion.
Line 216: "neonates who aged less than 60 days" - these are infants, not neonates.
RESPONSE: This was corrected.
I would suggest adding a paragraph about the limitations of the study, e.c. urine collection method, a small number of identified Gram-positive bacteria, etc.
RESPONSE: Limitations were added to the discussion.
English editing is required. For example:
- line 44 - "as they growing"
- lines 89-80 - "important is infected neonates"
- lines 100-102- should be rephrased
RESPONSE: These were corrected.
References: most of the cited articles were published within the last 20 years, however, the authors cite articles referring to urinary tract infections in infants published in 1993, 1993, and 1982. I would suggest looking for more recently published articles.
RESPONSE: these references were updated.